# Relationship Between Body Mass Index and Fat Mass Percentage with Proprioception in Children

**DOI:** 10.3390/jfmk10010059

**Published:** 2025-02-09

**Authors:** Eduardo Guzmán-Muñoz, Yeny Concha-Cisternas, Guillermo Méndez-Rebolledo, Manuel Vásquez-Muñoz, Pablo Valdés-Badilla, Tomás Herrera-Valenzuela, Cristian Núñez-Espinosa, Jordan Hernández-Martínez

**Affiliations:** 1Escuela de Kinesiología, Facultad de Salud, Universidad Santo Tomás, Talca 3460000, Chile; 2Pedagogía en Educación Física, Facultad de Educación, Universidad Autónoma de Chile, Talca 3460000, Chile; 3Vicerrectoría de Investigación e Innovación, Universidad Arturo Prat, Iquique 1100000, Chile; 4Department of Physical Education, Faculty of Education Sciences, University ofCádiz, 11519 Puerto Real, Spain; 5Centro de Observación y Análisis de Datos en Salud, Facultad de Medicina y Ciencias de la Salud, Universidad Mayor, Santiago 8580745, Chile; 6Escuela de Medicina, Facultad de Medicina y Ciencias de la Salud, Universidad Mayor, Santiago 8580745, Chile; 7Departament of Physical Activity Sciences, Faculty of Education Sciences, Universidad Católica del Maule, Talca 3460000, Chile; 8Sports Coach Career, School of Education, Universidad Viña del Mar, Viña del Mar 2200055, Chile; 9Department of Physical Activity, Sports and Health Sciences, Faculty of Medical Sciences, Universidad de Santiago de Chile (USACH), Santiago 8370003, Chile; 10Escuela de Medicina, Universidad de Magallanes, Punta Arenas 6200000, Chile; 11Centro Asistencial Docente e Investigación, Universidad de Magallanes, Punta Arenas 6200000, Chile; 12Department of Physical Activity Sciences, Universidad de Los Lagos, Osorno 5290000, Chile; 13Programa de Investigación en Deporte, Sociedad y Buen Vivir, Universidad de los Lagos, Osorno 5290000, Chile

**Keywords:** childhood obesity, proprioception, body mass index, body fat percentage, sensorimotor function, upper limbs, lower limbs

## Abstract

Background/Objectives: Childhood obesity is linked to motor and sensorimotor impairments, including proprioceptive deficits. While research has predominantly focused on lower limb proprioception, less is known about the impact on upper limbs. This study investigated the relationship between body mass index, body fat percentage, and proprioception of children aged 11–12 years. Methods: A quantitative, correlational, observational design was employed. BMI was calculated from weight and height measurements, body fat percentage was assessed via bioelectrical impedance analysis, and proprioception was measured using an active repositioning test with inertial sensors in 44 children. Results: Significant correlations were found between BMI and positional errors in the shoulder (r = 0.64, *p* < 0.001), elbow (r = 0.36, *p* = 0.007), and knee (r = 0.42, *p* = 0.002). Regarding body fat percentage, significant correlations were observed with positional errors in the shoulder (r = 0.28, *p* = 0.031), elbow (r = 0.46, *p* < 0.001), and knee (r = 0.29, *p* = 0.030). Regression analysis showed that BMI and body fat percentage significantly predicted positional errors in the shoulder, elbow, and knee. In the shoulder joint, girls demonstrated lower positional errors compared to boys, influenced by both BMI (β = −1.36, *p* = 0.015) and body fat percentage (β = −3.00, *p* < 0.001). Conclusions: Higher BMI and body fat percentage are associated with shoulder, elbow, and knee joint proprioceptive deficits. Interventions targeting weight reduction and proprioceptive training may mitigate these deficits and promote sensorimotor function in children.

## 1. Introduction

In the 21st century, overweight and obesity are reaching epidemic levels in many developed and developing countries [1,2]. Overweight and obesity are defined as an abnormal and excessive accumulation of fat that can harm health, manifesting in increased body weight and volume [3]. According to the World Atlas of Obesity, it is projected that by 2035, 39% of children will be obese [4].

Extensive documentation supports the link between childhood obesity and an elevated risk of developing chronic diseases, particularly cardiovascular disease, type 2 diabetes, and specific types of cancer later in life [5,6,7]. Functionally, childhood obesity can negatively affect various activities of daily life, including balance, stability, and locomotion, increasing predisposition to injury and fall risk [8,9,10]. Obesity also appears to contribute to reduced efficiency in performing motor action in a bipedal posture, likely due to the limitations in postural control imposed by excess weight [11,12,13]. Several studies have shown deficits in fine and gross motor skills development among obese individuals, encompassing walking, running, and jumping [14,15,16].

One of the causes proposed to explain the reported changes in motor skills and motor control in obese children corresponds to the alteration of the sensorimotor system [14,17,18]. The system includes all the afferent, efferent, and central integration components essential for maintaining the functional stability of joints [19]. It processes external and perceives internal sensory information and generates adequate motor responses [19]. Guzmán-Muñoz et al. (2024) reported a longer reaction time in the quadriceps muscle among obese individual compared to their normal-weight peers [20]. This delay highlights the detrimental impact of obesity on motor control, specifically in the integration and execution of neuromuscular responses. Chronic accumulation of adipose tissue and intramuscular fat infiltration is postulated to increase levels of circulating pro-inflammatory cytokines, which may impair muscle function and performance by promoting protein breakdown in muscles [21,22]. In addition, the accumulation of fat mass is associated with a slowdown in motor nerve conduction velocity [23,24], which could contribute to the neuromuscular changes observed in these children.

At the sensory level, studies have shown that children with obesity have a proprioception deficit compared to their normal-weight peers [25,26]. Proprioception is a continuous and unconscious sensory flow from muscles, tendons, joints, and skin, allowing muscle tension, balance, and movement control [27,28]. Two components are essential for an effective motor stabilization strategy: the ability to sense the position of a joint and the ability to perceive the body’s movement and its parts [27]. The proprioceptive information comes primarily from muscle spindles, Golgi tendon organs, cutaneous receptors, and capsular mechanoreceptors [28]. In children, the literature has focused specifically on the knee [25,26] and ankle [25] joints, finding a significant proprioceptive deficit in the knee. Despite these results, the relationship between obesity and proprioceptive deficits has only been explored in lower limb joints, and the changes that could be generated in upper limb joints are unknown. Likewise, in these studies, children were classified using BMI [25,26]. However, some authors point out that the body fat percentage would be a more detailed and accurate measurement of body composition to determine the negative effects of obesity [29,30].

This study, therefore, aimed to examine the relationship between BMI and body fat percentage and proprioception in the shoulder, elbow, hip, and knee joints in children. It is hypothesized that children with a higher BMI and body fat percentage will demonstrate a greater proprioceptive deficit in the joints evaluated.

## 2. Materials and Methods

### 2.1. Study Design

In this study, we employed a quantitative approach, correlational type, and cross-sectional design. The participants were assessed during a 15-min session in a room set at 21 °C, accompanied by their parents and/or guardians. During the tests, participants wore shorts and were barefoot. Measurements included BMI, body fat percentage, and proprioception.

### 2.2. Participants

The sampling method used in this research was non-probabilistic, based on convenience. The inclusion criteria included: (i) schoolchildren from a public school located in Talca (Chile); (ii) participants were aged between 11 and 12 years. The exclusion criteria included: (i) individuals with neurological disorders, (ii) those with musculoskeletal injuries in the upper and lower limbs such as fractures, sprains, dislocations, or muscle tears within six months before the assessments, (iii) the presence of any inflammatory or painful conditions affecting the upper and lower limbs at the time of the evaluations, and (iv) reliance on assistive devices for walking.

This study involved a total of 44 schoolchildren, consisting of 24 girls and 20 boys. The girls had an average age of 11.58 ± 0.44 years, a body mass of 48.05 ± 11.27 kg, and an average height of 1.51 ± 0.05 m. The boys had an average age of 11.58 ± 0.41 years, a body mass of 44.92 ± 8.39 kg, and a height of 1.42 ± 0.05 m. In accordance with the principles outlined in the Declaration of Helsinki, informed consent was obtained from both the participants and their parents through signed consent forms. This study was approved by the local Ethics Committee of Universidad Santo Tomás, Chile, under registration number 13320.

The sample size was determined using GPower software (Version 3.1.9.6, Franz Faul, Universiät Kiel, Germany) with the multiple linear regression statistical model. For this calculation, an alpha error of 0.05, a power of 0.9, and a number of 2 predictors are considered. The minimum sample size obtained for this study was 36 participants.

### 2.3. Body Mass Index (BMI)

During the assessments, participants were instructed to wear light clothing (shorts, a light t-shirt, and no shoes) to accurately measure their body weight and standing height. Body weight was recorded using a digital scale Omron Karada HBF-375 (Omron Corporation, Kyoto, Japan; accuracy of 0.1 kg), and height was measured with a stadiometer Seca model 220 (Seca, Hamburg, Germany; accuracy of 0.1 cm). BMI was then calculated by dividing the body weight in kilograms by the square of the height in meters (kg/m²).

### 2.4. Body Fat Percentage

Body fat percentage was measured using bioelectrical impedance analysis with the Omron HBF-375 body fat analyzer (Omron HBF-375 Karada Scan; Omron, Kyoto, Japan). This method was chosen due to its validated effectiveness and its frequent use in epidemiological studies for assessing body fat percentage in children [31]. For the measurement, the subject stood upright on the device base, holding the integrated handheld electrodes. The current passed between the foot and hand electrodes, enabling a full-body analysis that estimated body fat percentage. Measurements were conducted under standardized conditions, including 6–8 h of fasting and abstaining from fluid intake or intense physical activity before the assessment [32]. Body fat percentages were classified according to percentiles for sex and age, as defined by McCarthy et al. (2006): normal (2nd–85th percentile), overfat (>85th–95th percentile), and obese (>95th percentile) [33].

### 2.5. Proprioception

To assess proprioception, it was evaluated using the active repositioning test [25,26,34]. The relative positional error (°) was measured in the shoulder, elbow, hip, and knee joints through an isoinertial measurement unit (IMU). The IMU used for this evaluation was the Trigno Research+ system (Delsys, Boston, USA), with data obtained using the EMGWorks 4.9 software (Delsys, Boston, MA, USA). For each joint, the procedures described below were followed. The subject was placed in a comfortable position to minimize movement and with eyes closed to eliminate visual feedback. An assessor passively moved the joint to a “target” (reference) position and held it for 5 s [25,26,34]. The participant was asked to remember the position of the evaluated segment. The target positions for the joints included 70° of shoulder flexion, 80° of elbow flexion, 70° of hip flexion, and 50° of knee flexion [34]. According to the literature, selecting mid-range joint positions as targets is recommended, as these positions are reported to be more challenging to replicate compared to extremes of the range of motion [35]. Subsequently, the joint was returned passively to a neutral position, and the participant was asked to actively replicate the target position. When the participants indicated that they were in the reference position, the joint angle was recorded and compared to the exposed angle in the target position. The difference between these two values corresponded to the “relative positional error”. The joint evaluations were performed three times, and the average of the three measurements was considered [25,26,34].

### 2.6. Statistical Analysis

Data were analyzed using GraphPad Prism 9.0 statistical software (GraphPad Software, La Jolla, CA, USA). Descriptive statistics, including the mean and standard deviation, were calculated to summarize the sample’s characteristics: age, weight, height, BMI, fat mass, and relative positional error. The Shapiro-Wilk test was performed to assess data distribution. Since the data followed a normal distribution, Pearson’s correlation test was applied to examine the relationships between BMI and relative positional error, as well as between body fat percentage and relative positional error. A correlation coefficient of from 0 to 0.4 was considered weak, from 0.4 to 0.7 was moderate, and from 0.7 to 1.0 was strong. To determine the influence of gender on the results, multiple linear regression models (95% confidence interval) were performed, where the dependent variable was the relative positional error, and the independent variables were the BMI and body fat percentage adjusted for sex. For this analysis, nutritional status (normal weight, overweight, and obese), body fat percentage (normal, overfat, and obese), and sex (boys/girls) variables were categorized. The goodness of fit was determined using the R2 coefficient. A collinearity diagnosis was made for each variable in the regression models obtained, where variables with values less than 0.10 tolerance and values above 10.0 variance inflation factor (VIF) were eliminated. Statistical significance was set at *p* < 0.05 for all analyses.

## 3. Results

The sample evaluated obtained an average BMI of 21.93 ± 3.95 kg/m^2^, while the mean body fat percentage was 25.18 ± 11.58. Based on BMI, 40.9% of the participants were classified as normal weight, 27.3% as overweight, and 31.8% as obese. On the other hand, based on the body fat percentage, 54.5% were classified as normal, 9.1% as overfat, and 36.4% as obese. Table 1 shows the relative positional error results in the shoulder, elbow, hip, and knee joints.

Figure 1 shows the results of the correlations between BMI and positional error of the shoulder, elbow, hip, and knee joints. A significant correlation can be observed between BMI and positional error of the shoulder (*p* < 0.001; r = 0.64), elbow (*p* = 0.007; r = 0.36), and knee (*p* = 0.002; r = 0.42). The hip joint had no significant relationship (*p* = 0.226; r = 0.11).

Regarding body fat percentage, a significant correlation was observed with positional error of shoulder (*p* = 0.031; r = 0.28), elbow (*p* < 0.001; r = 0.46), and knee (*p* = 0.030; r = 0.29) (Figure 2). The hip joint had no significant relationship (*p* = 0.449; r = −0.02).

Table 2 shows the multiple linear regression models obtained for relative positional error time based on BMI. Significant results were found for the shoulder (R² = 0.37; *p* < 0.001), elbow (R² = 0.19; *p* = 0.013), and knee (R² = 0.14; *p* = 0.045) joints. The results indicate that BMI positively correlates with increased relative positional error time in these joints. Particularly, sex significantly influenced results only in the shoulder joint, with girls showing reduced error times compared to boys (β = −1.36; *p* = 0.015).

Table 3 presents the multiple linear regression models for relative positional error time based on body fat percentage. Significant findings were observed for the shoulder (R² = 0.35; *p* < 0.001), elbow (R² = 0.24; *p* = 0.003), and knee (R² = 0.11; *p* = 0.047) joints. The results indicate that increased body fat percentage correlates with higher relative positional error time in these joints. In the shoulder joint, girls again showed reduced error times compared to boys (β = −3.00; *p* < 0.001). No significant association was found between fat mass percentage and positional error time in the hip joint (R² = 0.03; *p* = ns).

## 4. Discussion

The findings of this research indicate that BMI and body fat percentage are correlated with performance in the proprioceptive positional error test in children. Specifically, the results show that higher BMI and a greater body fat percentage are associated with higher positional error values in the shoulder, elbow, and knee joints, indicating lower proprioceptive accuracy. When adjusting these results by sex through multiple linear regression, it was determined that only in the shoulder joint did sex influence the positional error results for both BMI and body fat percentage, with girls demonstrating better proprioception than boys. Previous studies have also reported similar findings, noting poorer proprioception in the knee joint in children with obesity [25,26]. In obese adults, a proprioceptive deficit has also been observed, particularly in the trunk joint [36]. However, unlike our study, these investigations did not include analyses of upper limb joints or consider the sex factor.

One possible explanation for the diminished proprioception observed in children with obesity is the increased body mass, which places additional load on the joints, particularly the knees and ankles [25]. This excess weight can compromise proprioceptive feedback, as evidenced by studies demonstrating a significant decline in knee joint proprioception among obese children, characterized by greater active repositioning errors compared to their normal-weight counterparts [25,26]. Excessive joint loading may exceed the functional capacity of proprioceptors to effectively convey information about joint position and movement sense, potentially exacerbating proprioceptive deficits. However, this hypothesis does not fully account for proprioceptive impairments observed in the upper limbs, such as the shoulder and elbow, which are not subject to direct weight-bearing loads.

An alternative and increasingly supported theory attributes these deficits to systemic inflammation associated with excessive adipose tissue accumulation, as seen in obesity. Adipose tissue, now recognized as a dynamic endocrine and immunological organ, secretes pro-inflammatory cytokines such as tumor necrosis factor-alpha (TNF-α) and interleukin-6 (IL-6) [17]. These mediators disrupt homeostasis by influencing both the central and peripheral nervous systems. Chronic exposure to these cytokines induces oxidative stress and activates inflammatory pathways, including nuclear factor-kappa B (NF-κB) signaling. Such molecular disruptions can impair the function of mechanoreceptors and compromise the structural and functional integrity of nerve fibers that transmit proprioceptive signals [17]. The resulting deficits in sensory feedback from the upper limbs may thus stem from these systemic inflammatory processes rather than direct mechanical overload, highlighting the multifactorial impact of obesity on proprioceptive function.

In this context, another plausible explanation is that the accumulation of excess fat within the muscles and around the joints could disrupt the standard mechanisms of sensory and motor responses. Studies have shown that obese people have a decrease in nerve conduction velocity [23,24,37]. The main cause attributed to this finding is based on nerve compression that causes fat accumulation [38]. Specifically, it has been reported that there is a deceleration of the nerve conduction velocity of the motor nerves and a reduction in the nerve action potential of sensory nerves in obese people compared to healthy subjects [38], which could impact neuromuscular control and proprioception, respectively. Also, the skin stretching from excess adiposity might increase the distance between cutaneous mechanoreceptors, potentially lowering the somatosensory discrimination threshold [39].

Another notable finding of this research was that girls exhibited better proprioception compared to boys, which contrasts with the findings of Das-Yadav et al. (2020), who reported poorer proprioception in adult females compared to adult males. However, in our study, the maturation process in children could be a key factor explaining these results [40]. Sex hormones, particularly estrogen, play a crucial role in the maturation of sensory and neuromuscular systems. Estrogen, the predominant hormone in girls, influences various aspects of the central nervous system by enhancing neuronal plasticity, improving neuromuscular communication efficiency, and optimizing the function of mechanoreceptors, which are responsible for perceiving joint position and movement sense [41,42]. Mosavi-Ghomi et al. (2021) reported that higher levels of estrogen could affect brain areas related to proprioceptive integration and cognition, which in turn may help reduce positional error [43]. On the other hand, it is hypothesized that estrogen has a significant impact on the neurotransmitter gamma-aminobutyric acid (GABA) in the hippocampus, enhancing its balanced inhibitory effect on brain signals. Proper inhibition mediated by GABA is crucial for neuronal modulation, which could improve the accuracy with which the brain processes and integrates proprioceptive signals [44].

On the other hand, our findings showed the absence of a significant correlation between BMI/body fat percentage and hip proprioception. This result may be due to the biomechanical characteristics of this joint. Unlike the knee, which is more exposed to weight-bearing forces and dynamic instability, the hip has a deeper socket and greater structural stability, which may make it less susceptible to proprioceptive impairments related to obesity [45].

The findings of this study emphasize the importance of considering proprioception as a key factor in managing childhood obesity. Given the relationship between increased BMI/body fat percentage and impaired proprioceptive performance, interventions targeting proprioceptive training could play a crucial role in mitigating these deficits. Specifically, balance exercises, joint stabilization activities, and neuromuscular coordination training should be incorporated into physical activity programs aimed at children with obesity to improve proprioceptive accuracy and reduce injury risk. Furthermore, integrating anti-inflammatory dietary strategies and promoting physical activity may help attenuate the systemic inflammation associated with adiposity, which could further support proprioceptive function. Finally, future longitudinal studies focusing on diverse pediatric populations are needed to confirm causal relationships and refine intervention strategies based on these findings.

Among the limitations of this study is its cross-sectional design, which prevents establishing causal relationships between obesity and the observed proprioceptive deficits. Additionally, the sample used, although representative of a specific group, was small and conveniently selected, limiting the generalization of the findings to other pediatric populations. Similarly, complementary biomechanical analyses, such as the assessment of muscle strength or motor control, were not included, nor were additional contextual factors, such as physical activity level, diet, or family history, considered, all of which could have influenced the results. Studies have revealed that these factors can cause alterations in body sensation [46]. For example, it has been seen that individuals with a higher level of physical activity have better body perception [46].

The main strength of this study lies in the use of validated tools such as bioimpedance analysis for estimating body fat percentage and an inertial measurement system (IMU) to assess proprioception, ensuring accuracy and reliability in the data. The focus on a specific population of children aged from 11 to 12 years, a critical stage for motor and proprioceptive development, allows for the generation of relevant findings for early interventions. Additionally, the multivariable analysis adjusted for sex provides the opportunity to identify significant differences in proprioception between boys and girls, considering biological and hormonal factors that may influence the results. Finally, this study’s findings have high practical application potential, as they can guide the design of clinical interventions and educational programs aimed at improving proprioception and mitigating the negative effects of obesity in the pediatric population.

## 5. Conclusions

BMI and body fat percentage are significantly correlated with proprioceptive performance in children, as higher values are associated with greater positional errors in the shoulder, elbow, and knee joints, indicating reduced proprioceptive accuracy. Sex-adjusted analyses revealed that sex only influenced proprioceptive performance in the shoulder joint, where girls demonstrated better accuracy than boys. Based on these results, interventions for children with overweight and obese should extend beyond weight reduction to include training and rehabilitation programs that enhance proprioceptive accuracy and promote motor development. Programs tailored to improve body composition by decreasing body fat percentage and addressing systemic inflammation may help preserve sensorimotor function and reduce the impact of obesity on joint performance. Such integrative approaches could mitigate proprioception deficits, safeguard motor competence, and support the overall physical development of pediatric populations.

## Figures and Tables

**Figure 1 jfmk-10-00059-f001:**
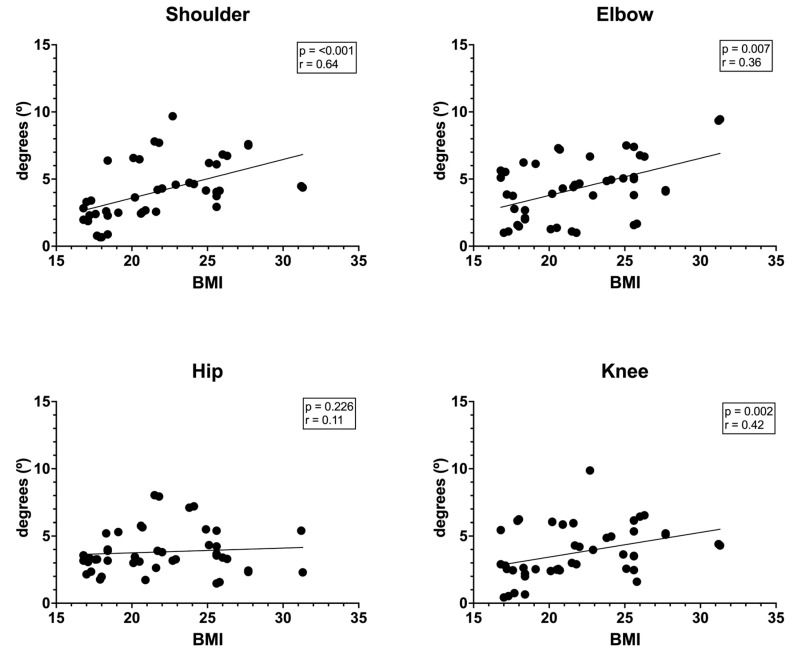
Correlation between BMI and relative positional error.

**Figure 2 jfmk-10-00059-f002:**
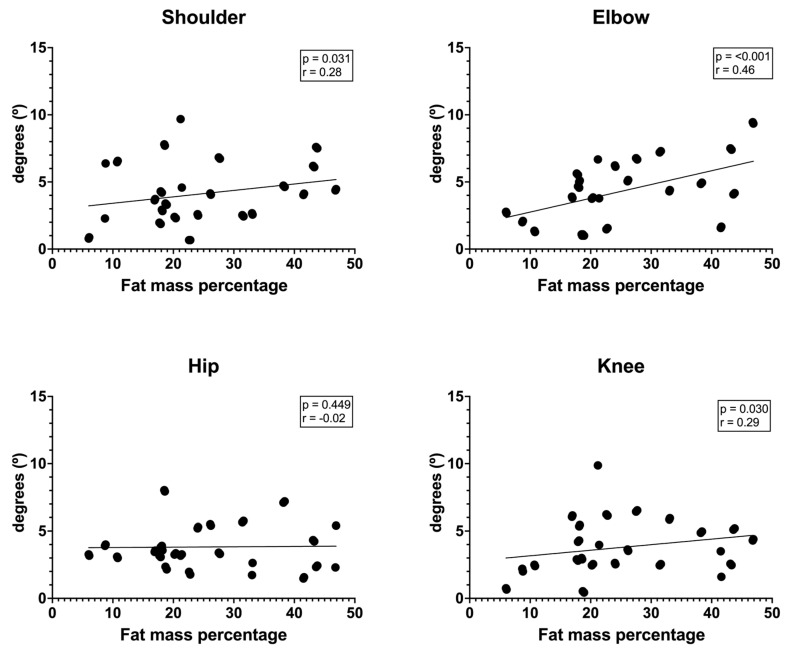
Correlation between body fat percentage and relative positional error.

**Table 1 jfmk-10-00059-t001:** Relative positional error in the shoulder, elbow, hip, and knee joints.

Joint	Mean ± SD	Min-Max
Shoulder (degrees)	4.34 ± 2.33	0.67–9.68
Elbow (degrees)	4.66 ± 2.88	1.00–12.78
Hip (degrees)	3.73 ± 1.66	1.47–8.03
Knee (degrees)	3.88 ± 2.21	0.43–9.97

SD: standard deviation.

**Table 2 jfmk-10-00059-t002:** Multiple linear regression models obtained for relative positional error time according to BMI.

Joint	R^2^	Coefficient β	*p value*	95% CI	
Shoulder	0.37		<0.001		
Intercept		−1.87	<0.001	−4.97 to −1.22	
BMI		0.30	<0.001	0.16 to 0.44	
Girls (ref. boys)		−1.36	0.015	−2.46 to −0.27	
Elbow	0.19		0.013		
Intercept		−1.84	ns	−6.01 to 2.32	
BMI		0.27	0.004	0.09 to 0.46	
Girls (ref. boys)		0.34	ns	−1.13 to 1.81	
Hip	0.03		ns		
Intercept		3.14	0.033	0.25 to 6.04	
BMI		0.04	ns	−0.08 to 0.173	
Girls (ref. boys)		−0.49	ns	−1.51 to 0.52	
Knee	0.14		0.045		
Intercept		−0.17	ns	−3.43 to 3.07	
BMI		0.18	0.013	0.04 to 0.33	
Girls (ref. boys)		−0.25	ns	−1.40 to 0.89	

95% CI: 95% confidence interval; ns: no significant.

**Table 3 jfmk-10-00059-t003:** Multiple linear regression models obtained for relative positional error time according to body fat percentage.

Joint	R^2^	Coefficient β	*p value*	95% CI	
Shoulder	0.35		<0.001		
Intercept		2.47	<0,001	1.13 to 3.82	
Fat percentage		0.13	<0,001	0.06 to 0.19	
Girls (ref. boys)		−3.00	<0,001	−4.43 to −1.56	
Elbow	0.24		0.003		
Intercept		1.75	0.047	0.02 to 3.48	
Fat percentage		0.13	0.001	0,05 to 0.21	
Girls (ref. boys)		−1.42	ns	−3.26 to 0.42	
Hip	0.03		ns		
Intercept		3.62	<0.001	2.30 to 4.85	
Fat percentage		0.02	ns	−0.03 to 0.08	
Girls (ref. boys)		−0.83	ns	−2.14 to 0.47	
Knee	0.11		0.047		
Intercept		2.57	<0.001	1.15 to 3.98	
Fat percentage		0.07	0.028	0.01 to 0.13	
Girls (ref. boys)		−1.16	ns	−2.67 to 0.34	

95%. CI: 95% confidence interval; ns: no significant.

## Data Availability

The datasets used, and the data analyzed in this study will be made available upon reasonable request to the corresponding author (E.G.-M.).

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
