# Peer review of "Relationship Between Body Mass Index and Fat Mass Percentage with Proprioception in Children"

_jfmk, 2025, doi:10.3390/jfmk10010059_

Round 1
Reviewer 1 Report
Comments and Suggestions for Authors
Many thanks to the Editor for the opportunity to revise the article entitled “Relationship Between Body Mass Index and Fat Mass Percentage With Proprioception in Children”, in which the Authors have investigated the relationship between childhood obesity and upper/lower body proprioception, a very interesting and original topic. The research is well-written in its parts, with accurate explanations of diminished proprioception observed in children with obesity. However, I pointed out some issues that need further clarification, as follows:
May I ask You, why haven’t you expanded the research (i.e. data collection) in other ages (9-11 years old and 12-14 y.o.), considering that You had the possibility to collect data from a public school?
Why haven’t you also measured ankle and wrist positional error, as it was done in the paper you referenced [34] in the methodology section?
As correctly cited in the limitations section, why haven’t you considered the participants' physical activity level? Would you be able to collect it retrospectively? Because it is a very important confounding factor.
Have you checked if the isoinertial measurement unit used in this research reports accurate and valid data in measuring the relative positional error (°)?
Do you think a 3-5° error in the joint position is a minimally important difference in this kind of test? Or maybe it could lie in the measurement error of the device?
In Table 1, can I ask You to insert mean(SD) in a single column and, again, Min-Max in a single column? So, the resulting Table 1 would have only 3 columns instead of 5.
Why do you think hip positional error was unaffected by BMI and body fat percentage? Considering that the hip joint, along with the knee, must sustain most of the body weight.
Have you considered which joint had the most significant positional error? If not, why? It would not be interesting to know the more to less affected joints in order to propose, for example, a priority in the proprioception training? Could it fit in a future direction study?
Author Response
Comment 1: May I ask You, why haven’t you expanded the research (i.e. data collection) in other ages (9-11 years old and 12-14 y.o.), considering that You had the possibility to collect data from a public school?
Response 1: We acknowledge the importance of including a broader age range (e.g., 9–11 and 12–14 years old) in our study. However, our selection of children aged 11–12 years was based on two main considerations: (i) ensuring a homogeneous sample in terms of neurodevelopmental and motor maturation, as proprioceptive capabilities may vary significantly across different stages of childhood and adolescence, and (ii) logistical constraints regarding school approvals and parental consents, which limited our ability to include a broader age range. Future research should indeed explore these age groups to provide a more comprehensive understanding of the relationship between obesity and proprioception.
Comment 2: Why haven’t you also measured ankle and wrist positional error, as it was done in the paper you referenced [34] in the methodology section?
Response 2:
While we referenced previous studies that assessed proprioception in the ankle and wrist, our study focused primarily on joints that are highly involved in functional movements, such as the shoulder, elbow, hip, and knee. These joints are critical in activities of daily living and motor performance in children. Additionally, logistical constraints, including time limitations per participant and the number of assessments feasible within a school setting, led us to prioritize these joints. Nevertheless, assessing wrist and ankle proprioception in future studies would be valuable.
Comment 3: As correctly cited in the limitations section, why haven’t you considered the participants' physical activity level? Would you be able to collect it retrospectively? Because it is a very important confounding factor.
Response 3: We acknowledge that physical activity level is an important confounding factor that could influence proprioceptive performance. Unfortunately, we did not collect this data at the time of the study. Retrospective data collection poses challenges due to recall bias, especially in children. However, future studies should include objective assessments of physical activity, such as accelerometry or validated questionnaires, to strengthen the findings.
Comment 4: Have you checked if the isoinertial measurement unit used in this research reports accurate and valid data in measuring the relative positional error (°)?
Response 4: The IMU used in our study (Trigno Research+ system, Delsys, Boston, USA) has been validated in previous research for assessing joint angles and movement patterns. However, we acknowledge that the device’s accuracy in measuring proprioceptive errors may have intrinsic limitations.
Comment 5: Do you think a 3-5° error in the joint position is a minimally important difference in this kind of test? Or maybe it could lie in the measurement error of the device?
Response 5: 3–5° difference in joint position error is within the range observed in previous proprioception studies. While it may partly reflect measurement error, we believe it still represents meaningful proprioceptive variability, particularly considering the consistent associations observed with BMI and fat mass percentage. Further studies should determine a specific MID for proprioception assessments in pediatric populations.
Comment 6: In Table 1, can I ask You to insert mean(SD) in a single column and, again, Min-Max in a single column? So, the resulting Table 1 would have only 3 columns instead of 5.
Response 6: OK. We have modified it as you suggested. Thank you very much.
Comment 7: Why do you think hip positional error was unaffected by BMI and body fat percentage? Considering that the hip joint, along with the knee, must sustain most of the body weight.
Response 7: The absence of a significant correlation between BMI/body fat percentage and hip proprioception may be due to the biomechanical characteristics of this joint. Unlike the knee, which is more exposed to weight-bearing forces and dynamic instability, the hip has a deeper socket and greater structural stability, which may make it less susceptible to proprioceptive impairments related to obesity. We added this explication in the discussion.
Comment 8: Have you considered which joint had the most significant positional error? If not, why? It would not be interesting to know the more to less affected joints in order to propose, for example, a priority in the proprioception training? Could it fit in a future direction study?
Response 8: We agree that identifying the joint most affected by proprioceptive errors would provide valuable insights for designing targeted proprioception training interventions. Based on our findings, the shoulder exhibited the most significant proprioceptive deficit. Future studies should explore whether proprioceptive training interventions should prioritize certain joints over others to optimize motor function in children with obesity.
Reviewer 2 Report
Comments and Suggestions for Authors
This paper deepens the Relationship Between Body Mass Index and Fat Mass Percentage With Proprioception in Children.
After a careful analysis, many concerns have to be corrected or better addressed.
Firstly, the study design is not clearly presented. Is it a cross sectional sytudy? Please, specify this apsect.
The, the sample size calculation is missing and it is mandatory. A convenience sample, which anyway requires some leterature references with similar articles, does not allow you to make solid statistical anlysis and, as a result, valid conclusions.
The inclusion/exclusion criteria are not solid too. You said as follows: "The inclusion criteria included: (i) schoolchildren from a public school located in Talca (Chile); (ii) participants were aged between 11 and 12 years. The exclusion criteria included: (i) individuals with neurological disorders, (ii) those with musculoskeletal injuries in the lower limbs such as fractures, sprains, dislocations, or muscle tears within six months before the assessments, (iii) the presence of any inflammatory or painful conditions affecting the lower limbs at the time of the evaluations, and (iv) reliance on assistive devices for walking.". So, why did you limit the sample to children aged 11-12 years? Why did you exclude those suffering from musculoskeletal injuries and inflammatory or painful conditions in the lower limbs and not also in the upper ones? You investigated also shoulders and elbows.
Then, who did perform the proprioception evaluation? Did you consider the possible influence of sporting activities on these children skilss? If you did not investigate these aspects, please state why.
Discussion is interesting but too weak at the moment. In order to improve that according to the issues mentioned abouve, and also in order to gice your results a more clinical value, I suggest the following references:
-Notarnicola A, Farì G, Maccagnano G, Riondino A, Covelli I, Bianchi FP, Tafuri S, Piazzolla A, Moretti B. Teenagers’ perceptions of their scoliotic curves. an observational study of comparison between sports people and non- sports people. Muscles Ligaments Tendons J [Internet]. 2019;9(2):225-35.
Best regards and good luck
Author Response
Comment 1: Firstly, the study design is not clearly presented. Is it a cross sectional sytudy? Please, specify this apsect.
Response 1: We acknowledge that the study design was not clearly specified. This study follows a cross-sectional design, and we have now explicitly stated this in the methodology section to ensure clarity.
Comment 2: The, the sample size calculation is missing and it is mandatory. A convenience sample, which anyway requires some leterature references with similar articles, does not allow you to make solid statistical anlysis and, as a result, valid conclusions.
Comment 2: The truth is that we did perform the sample calculation, but I don't know why we didn't add it to the manuscript. We have added it in the new version.
Comment 3: The inclusion/exclusion criteria are not solid too. You said as follows: "The inclusion criteria included: (i) schoolchildren from a public school located in Talca (Chile); (ii) participants were aged between 11 and 12 years. The exclusion criteria included: (i) individuals with neurological disorders, (ii) those with musculoskeletal injuries in the lower limbs such as fractures, sprains, dislocations, or muscle tears within six months before the assessments, (iii) the presence of any inflammatory or painful conditions affecting the lower limbs at the time of the evaluations, and (iv) reliance on assistive devices for walking.". So, why did you limit the sample to children aged 11-12 years? Why did you exclude those suffering from musculoskeletal injuries and inflammatory or painful conditions in the lower limbs and not also in the upper ones? You investigated also shoulders and elbows.
Response 3: Thank you for the observation, and I would like to clarify a few points regarding the inclusion and exclusion criteria. The limitation of the sample to children aged 11 to 12 years was established to study a homogeneous group in terms of physical and cognitive development, as this age range represents a crucial period in the development of motor skills and functions related to physical activity. However, we understand that a more detailed justification of this choice may be beneficial to contextualize the study design.
Regarding the exclusion of participants with musculoskeletal injuries or inflammatory/painful conditions in the lower limbs, it is true that we also considered shoulders and elbows as exclusion criteria. However, this was a writing error, and in fact, all exclusion criteria should have been applied uniformly to both the upper and lower limbs. Consequently, children with musculoskeletal injuries or inflammatory/painful conditions in any of the limbs, both upper and lower, were excluded from the study to avoid potential biases in the results, as such conditions could affect physical performance and joint functionality.
Comment 4: Then, who did perform the proprioception evaluation? Did you consider the possible influence of sporting activities on these children skilss? If you did not investigate these aspects, please state why.
Response 4: The proprioception evaluation was conducted by trained professionals with expertise in neuromuscular assessment. However, we did not collect specific data on the children's participation in sporting activities, which could influence proprioceptive ability. This has been acknowledged as a limitation in the discussion, and we suggest future research should control for this variable.
Comment 5: Discussion is interesting but too weak at the moment. In order to improve that according to the issues mentioned abouve, and also in order to gice your results a more clinical value, I suggest the following references:
Response 5: We appreciate the suggestion to strengthen the discussion section. We have revised this section to provide a more robust interpretation of the findings, addressing study design limitations, sample characteristics, and potential implications for clinical practice. Additionally, we have incorporated the recommended reference.
Round 2
Reviewer 1 Report
Comments and Suggestions for Authors
Authors have solved every point of the revision, I have no further requests.
Reviewer 2 Report
Comments and Suggestions for Authors
Thank you for the efforts to improve your paper according to my suggestions.
Now it seems well structured and ready for publication in my opinion, so no further corrections are needed.
Regards